# Deconstructing celebratory acts following goal scoring among elite professional football players

Assaf Lev[1,2,3]*, Yair Galily[3,4], Omer Eldadi[3,4], Gershon Tenenbaum[3,4]

1 Department of Sports Therapy, Ono Academic College, Kiryat Ono, Israel, 2 Sammy Ofer School of Communications, Interdisciplinary Center, Herzliya, Israel, 3 Sport, Media and Society (SMS) Research Lab, Sammy Ofer School of Communications, Interdisciplinary Center, Herzliya, Israel, 4 Ivcher School of Psychology, Interdisciplinary Center, Herzliya, Israel

☯ These authors contributed equally to this work.
* anthropolev@gmail.com

**Data Availability Statement:** All relevant data are within the manuscript and its Supporting Information files.

**Funding:** The authors received no specific funding for this work.

## Abstract

Goal celebration in sport is mostly spontaneous and is manifested via vocal expressions and bodily gestures aimed at communicating emotions. The aim of this study is to deconstruct the celebratory acts among elite professional football players in the European Champions League following scoring a goal, and to capture the multiple acts and functions of the celebrations. In examining the 2018/19 season of the European Champions League tournament, we draw attention to the players' celebrations and their corresponding social and individual functions. All goals/celebrations ($K = 366$) were used for the analyses. To analyze the goal celebration acts, a socio-psychological model was established which is comprised of several theories. To describe the goal celebration acts across the competition stages (e.g., preliminary and final), match location (i.e., home or away), time phase (0–15, 15–45, 45–75, 75–90, 90+ minutes), scoring mode (i.e., prior to the goal, after the goal), and players' continent origin (Europe, Africa, Asia, South/Central, and North America), the number and percent of all the celebratory acts were counted and presented in their respective mode (i.e., single, double, and team). The main findings indicate that (a) most of the goal celebration acts were performed interactively by the scoring player and his teammates, (b) the interactive modes of celebration lasted longer than the modes which were performed non-interactively, (c) the celebration lasted longer following goal scoring in the final stage than in the preliminary stage, (d) the celebration duration lasted the longest time when the goal was scored during the overtime phase (90+ min) of the final but not the preliminary stage, and (e) players from Africa and South America demonstrated religious acts more than their European counterparts. We assert that our conceptual model enables the categorization of a variety of personal and social meanings to the celebrations on the field during the most thrilling moments of the game.

**Competing interests:** The authors have declared that no competing interests exist.

## Introduction

Goal celebration is the act of rejoicing at the scoring of a goal. Customarily performed by the scorer, the celebration may include his/her teammates, the team manager, the coaching staff, and/or the followers of the team. The term can refer to the celebration of a goal in general, as well as to specific actions such as a player removing his shirt, thanking God, or performing a somersault.

Frequently, the moments following the scoring of a goal in professional football (soccer) are filled with enthusiastic bodily gestures and unbridled joy. In 1982 FIFA (The Fédération Internationale de Football Association) issued guidelines that the 'exuberant outbursts of several players at once jumping on top of each other, kissing and embracing, should be banned from the football pitch'. However, according to Burnton [1] this suggestion didn't work. Players liked kissing each other so much that no amount of official declarations were going to stop them. In 1996 FIFA was forced to back down, revising their guidelines to allow 'reasonable celebration' but warning that 'choreographed celebrations' were not to be encouraged.

Immediately after scoring a goal the scorer becomes the center of the media's and spectators' attention. Utilizing the focus of the camera, the scoring player and his/her teammates celebrate their desired outcome. These precious seconds must be wisely capitalized upon to convey a message aimed at leveraging the player's career. The object of this study is to capture the multiple acts and functions of the celebrations following goal scoring. By analyzing the 2018/19 season of European Champions League games, we draw attention to the players' celebrations and the corresponding social and individual functions of these acts.

Celebration after scoring a goal is context dependent. It is usually expressed spontaneously and impulsively but has both individual and social-cultural roles [2]. Celebrations emphasize the loyalty and solidarity of the scoring player to the club for which he/she plays [3]. Several examples of behavior exhibiting solidarity and loyalty are hugging teammates, running to celebrate with fans, provoking the fans of the opposing team, kissing the team shirt's logo, and hugging the coach and/or the player's bench teammates. Celebrations are also aimed at promoting personal and social motives, such as strengthening one's personal elements by promoting bodily capital (e.g., exposing a muscular body by taking off the jersey), revealing an undershirt that contains a message, a religious ritual that praises and/or thanks God, and utilizing screen time for social protest [2]. Some celebrations comprise components of both themes while others focus on one single theme.

The study of post-performance behaviors in soccer is rare [3]. Among the few that do exist, Bornstein and Goldschmidt [4] imply that post-performance behaviors reflect the team's cohesion. Athletes in different sporting realms reported feeling basic emotions such as cheerfulness, enjoyment, and pride as a result of successful performances [5–7]. Expression of mood and emotions is transferable among teammates through a contagion process [3, 8, 9], which intensifies following goal scoring, leading to a celebration. The display of emotions following goal scoring affects the player through internal feedback loops [10], and results in cultivating and/or intensifying this emotional state [11]. The aim of this study is to sort out the post goal-scoring celebrations and provide an interpretation of these acts by referring to values such as *collectivity* and *individualism*.

To capture the fundamental underlying mechanism of the post-goal celebration and its consequences, we integrate ideas from several cognitive and emotion conceptual frameworks. In general, people interact with the environment via two systems, which Kahneman [12] terms *Fast* (system 1) and *Slow* (system 2). The fast system (system 1) relies on perception-action coupling and is intuitive–operating automatically and quickly with little or no perceptual-cognitive effort and without intentional or voluntary control. System 2, (slow), relies on

deliberative processes which allocate attention to the effortful mental activities that demand it. The two systems entail cognitive, emotional, and motor expressions in an interactive manner, which is illustrated graphically in Fig 1. These systems interact immediately and consequently to an event which triggers such a response, but is delayed when the response calls for an action or alteration. Celebration in sport is mostly spontaneous and repetitive, and is manifested via vocal and bodily gestures to express emotions. There are a number of studies on nonverbal behaviors in sport (e.g., [3, 13–21]), and for the most part (with the exception of [17] and [21]) not in the context of goal celebration. The celebration satisfies internal and personal needs and is rooted in cultural and contextual constraints. Specifically, scoring a goal is a stimulus which triggers the fast system to celebrate spontaneously by using vocal, facial, and bodily expressions (i.e., responses) to satisfy internal needs, and in cases, emphasize cultural roots [2]. However, one cannot ignore the role of the slow system in the celebration process. System 2 is very much related to specific appraisals of various emotions [22]. The significance of the goal depends on the player's interpretive process, but possibly is mediated by social interactions. In this vein, the celebration is a consequence of expected interactions with the player's teammates, coaching staff, bench-players, and spectators.

Goffman's [23] dramaturgical perspective is considered a critical cultural analysis [24, 25], and it provides a solid conceptual framework for players' goal celebrations. According to Goffman, social interaction, such as goal celebration, is a stage where the player strives to promote his/her honest and unbiased social identity. In this reality, the performer is a 'fabricator of impressions involved in the all-too-human task of staging a performance' [23, p. 252]. Specifically, the player is an actor who has *Front* and *Back regions of behavior*. The 'Front region of behavior' refers to the place the performance is given, where the actor engages in and performs his/her role for the audience, while the 'Back region of behavior' is considered a place where the performer can step out of character and cease to 'play a part' for an audience. Moreover, according to Goffman, to construct and maintain a trustworthy and credible impression as part of the Front region, one must be persistent in conveying expression in both the 'gives' and 'gives-off' channels. The 'gives' channel involves, for the most part, the *verbal symbols* used and conveyed purposefully, while the 'gives-off' channel involves a range of various actions transmitted via *nonverbal symbols* such as bodily gestures and props. The current paper centers on the players' celebrations ('gives-off') in front of the fans and cameras (i.e., the 'front region of behavior'). Detecting the players' performances on the football field via Goffman's theoretical perspective is useful, given the fact that:

> 'On the dramaturgical dimension, sport competition is defined as a play/spectacle, all actors acting towards putting the competition on stage in front of the public. The focus is on the performance act and on the way sport actors manage to respond to the social constraints and expectations attached to their role'

> [26, p. 508].

Though the back and front stage metaphors may illuminate the goal celebration phenomenon, it must also be considered in the context of personal and social psychological perspectives. Most theories of emotions consider the emotion response as one that aims at coping, overcoming and achieving personal objectives [e.g., 27–29]. However, emotions are determined largely by culturally supplied aims (see [30] for concrete examples). Specifically, most of the causes of emotional expressions are interpersonally, institutionally, and culturally defined.

Emotions provide information to the self [31] and influence others' cognition, attitudes, and behaviors [32]. The expression of emotions has consequences for other people, and serves

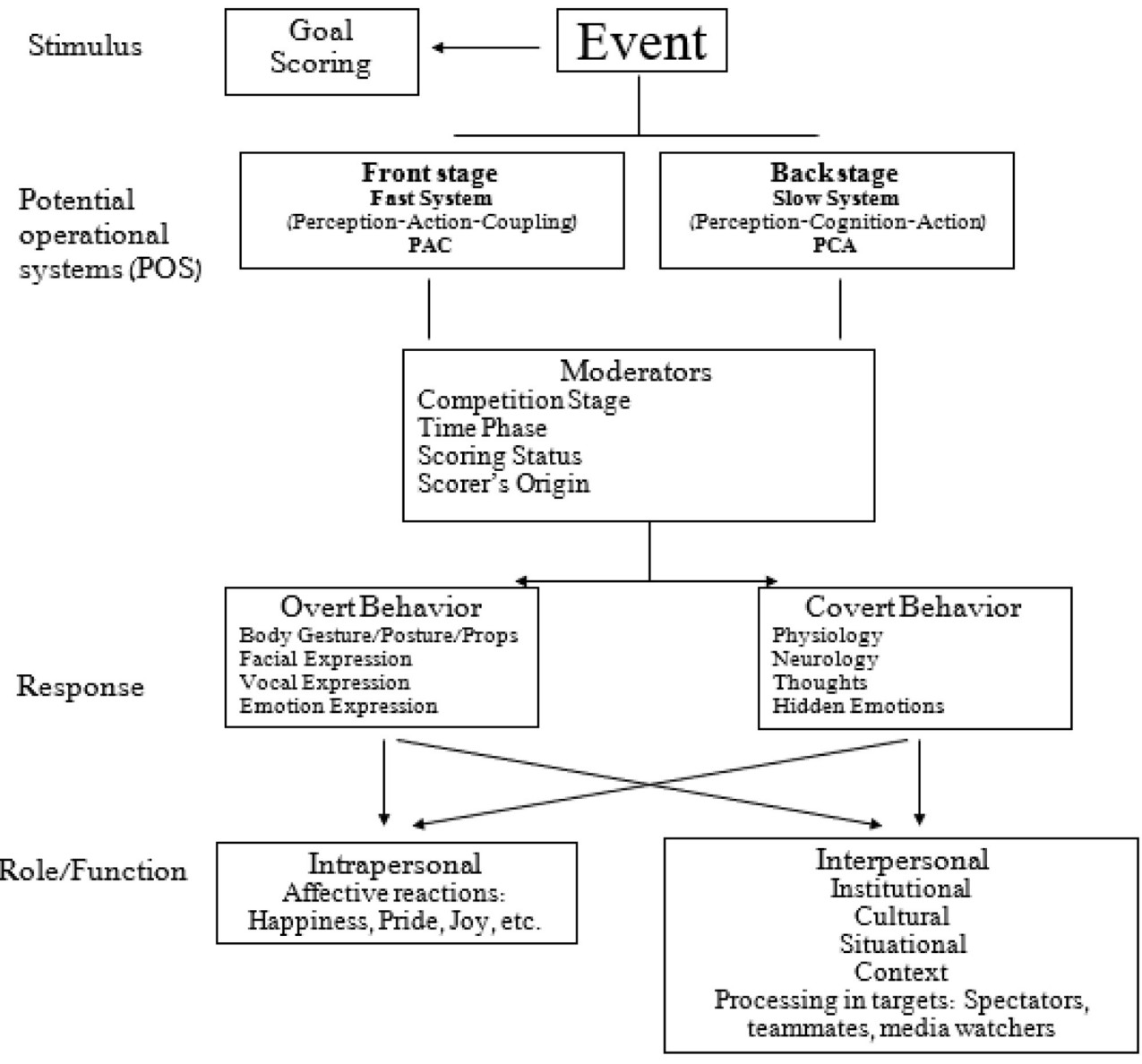

**Fig 1. A model depicting the goal celebration process, role, and expression.** The celebration is perceived as an overt expression of emotions mediated by environmental constraints and targeted toward interpersonal and intrapersonal roles.

interpersonal and cultural functions along with being internal and reactive phenomena (see [3, 33] for review). According to Parkinson, besides the pre-wired neurological-physiological expressions of emotions, emotional expressions are supplemented and supplanted by social and psychological interpretations, elaborations, and consequences. Moreover, cultures and institutions promote implicit and explicit expectations about emotional expressions. Thus, societal, cultural, and institutional constraints feed directly into the process of determining emotions in the first place [33].

Moreover, relying on Van Kleef's Emotions as a Social Information (EASI) model [2], Van Kleef, Van Doorn, Heerdink, and Koning [32] claimed that the primary function of emotions at the personal level is engendered by *social influence*. Accordingly, emotional expressions

following the goal scoring produce interpersonal effects by triggering affective reactions (e.g., joy, happiness, pride) and inferential processes in 'targets' (e.g., spectators, teammates, rivals, media watchers, etc.). Within the context of soccer playing, the extent of the target's reaction depends on the *information processing depth* and the *perceived appropriateness* of the emotional expressions. Thus, according to the EASI model, emotional expressions maintain personal functions required for adaptation and action readiness [28, 34] but also share a social function which engenders social influence. Van Kleef et al. [32] consider the interpersonal effect of emotions in social encounters such as negotiations, leadership, attitude change, compliance with requests, and conformity. In this context, we view goal celebration as an event of social-cultural function beyond the personal expression of joy, happiness, and the exhibition of superiority over the rival team. In line with Kahneman's [12] information processing systems, the expression of joy and happiness is spontaneous and fast, while the social functions are deliberate and, in most cases, pre-planned. The two systems feed on each other within the celebration event (see model in Fig 1).

The expression of emotions by players immediately after scoring a goal is a form of communication, in which evaluative representations are made to other people. The response is reflected by physiological-neurological and cognitive outcomes, but the syndrome is driven by social considerations [33]. For example, the majority (three-fourths) of the emotional reactions of people were attributed to the persons' relationship with other people [35], thus it is conceivable to believe that goal celebration is a personal expression of joy that is also driven by and aims at expressing social messages to others (e.g., spectators, teammates, rivals, non-attendees, etc.). This paper is designed to observe and analyze goal scoring celebration acts (see Fig 1) in 125 matches played in the European Premier League (EPL) during the 2019 season, and to measure their duration, categorize their meaning, nature, and message (i.e., interpersonally, institutionally, and culturally), and contrast these characteristics in the preliminary and final stages of the competition. Moreover, we also consider the effect of time phase during the competitive match, score status before and after the goal, the scoring team (home and away), and the continent origin of the scorer in describing the goal celebration. Overall, the aim is to operationally define the goal celebration characteristics within a more general value system which incorporates *collectivity* and *individualism*.

## Method

### The Champions League (CL) framework

The European Football Cup (now the *Champions League*) is a seasonal football tournament, established in 1956. The league is open to all 55 UEFA (the Union of European Football Associations) Champions Leagues, as well as teams that finished second through fourth in the Senior Leagues. Prior to the 1992/93 season, the tournament was called the 'European Cup'. In 1997, teams that finished in second place were also allowed to take part in the competition. The 1999/2000 season ushered in a less stringency system under the terms of the Champions League entry, with the first three UEFA leagues (Spanish League, Italian Serie A, and German Bundesliga) sending four teams each, while the next three in the rankings (Premier League, French League 1, and the Dutch Premier League) sent three delegates each.

According to Frick [36], since 1960, increasing numbers of football players from Eastern Europe, South America, Africa, and Asia have migrated to the top leagues in Western Europe (England, France, Germany, Italy, and Spain). This phenomenon has been massively fostered by the Bosman ruling. This ruling, or Bosman Law, is an act that changed the face of professional football in Europe (and in many respects the world), stating that players may leave their teams at the end of the contract without compensation, and invalidated the restrictions that

prevented teams from sharing more than a certain number of non-local players. As a result of the ruling, issued in late 1995, there were a number of significant changes in European football, most notably the many players from the EU who played outside their country in one of the other EU countries. There has also been a sharp increase in player salaries and contract extensions.

The UEFA Champions League final is the most watched annual sporting event in the world. The final of the 2012/13 season, for example, which featured Borussia Dortmund vs Bayern Munich, was aired in 203 countries and had the highest TV rating of the year worldwide with 360 million viewers. According to UEFA.com, the 2018/19 Final (Liverpool vs Tottenham) attracted 358 million viewers with a season that included players from 66 different nationalities (e.g., 62 from Brazil, 61 from Spain, 55 from France, and 47 from Germany).

## Scenarios' selection

In this study, we observed matches played from late June 2018 till May 2019, with 79 teams (from 54 European associations) taking part in the opening round, three qualifying rounds, and a play-off round; all qualifying rounds (except the preliminary round) are played under a two-legged knockout system (home and away basis).

The six remaining teams entered the group stage, joining 26 teams that qualified in advance. The 32 teams were drawn into eight groups of four teams and played each other in a double round-robin system. Subsequently, 32 teams participated in the UEFA Champions League. To represent the Champions League goal celebration phenomenon, all the 125 matches played in the Premier League during the 2018/19 season were recorded and all the goals/celebrations ($K = 366$) were used for the analyses. As stated, we sampled all 366 goals scored at the Champions League (exclusive of qualification sequences) from 125 games played. There were no selection criteria, and thus no selection bias: Every UEFA Champions League begins with a double (round-robin) group stage of 32 teams, which since the 2009/10 season is preceded by two qualification sequences for teams (N = 79 in the 2018/9 season) that do not receive direct entry to the tournament. The celebration acts in several analyses were broken down by competition stage–*preliminary (group) stage* (91 matches) and *final (knockout–top 16) stage* (27 matches), along with time phase in the competitive match, scoring status before and after the goal, and the goal scorer's origin.

## Coding celebratory behaviors, quality, and trustworthiness

According to Dael, Mortillaro, and Scherer [37] there are several methods of coding body movement in nonverbal behavior research, but they claim there is no consensus on a reliable coding system that can be used for the study of emotion expression. They proposed a new integrative method, which they termed 'the body action and posture coding system', which comprehensively demonstrated intercoder reliability at three different levels (occurrence, temporal precision, segmentation). Of the three dimensions they proposed for the body action and posture coding system (anatomical, form, and function), we relied mainly on observing the form of the goal celebration acts, but assigned their functions to individual and social roles driven from a sound theoretical foundation, which led to a newly established model that guided our study.

Video clips of the celebration acts were titled immediately after each observation by one of the authors, who has expertise in body expression. This author, who performed the coding, has extensive formal education and experience in teaching, coding, and interpreting body motions of artists, politicians, and instructors. To finalize the coding scheme, a negotiated approach was taken wherein the main coder and the other authors actively discussed the identified codes, while aiming to form a final version with the entire coded celebratory actions

[38]. The discussions resulted in a final full agreement about the coded behaviors. Nevertheless, it should be acknowledged that a limitation of the study was that it utilized an ad-hoc coding procedure and there may not be consensus as to its reliability. All the coded behaviors expressed positive emotions of joy, pride, and happiness. However, the coding system did not allow for distinguishing among the designated emotions. The coder observed each act and assigned to it a code and a title. Following the observation of 366 goals, the final celebration acts were condensed by all the authors into coherent categories. The actions were divided into three modes: (1) single mode, which consisted of 14 possible actions performed by the scoring player, (2) double mode, where two players celebrated the goal together, which consisted of four possible actions, and (3) team mode, where more than two players joined the celebration, which consisted of four possible actions. The total number of celebration acts ($N = 1190$) was divided into 14 single acts ($N = 731$, 61.4%), four double acts ($N = 114$, 9.6%) and four team acts ($N = 345$, 29.0%). The number and percentage of goal celebrations in 366 games and possible 22 celebration modes are presented in Table 1. (All graphics in this paper are original and drawn by the artist, Yehuda Hillman, for this study to demonstrate the celebration acts described above and do not represent any specific individual player or team. The artist waives all rights of artistic ownership and/or royalties for the graphics included in this paper).

To ensure quality and trustworthiness, we followed Tracy's [39] conceptualization of quality. Our study focused on a new topic with meaningful implications for society, given the significant role soccer plays in society worldwide. It also holds significant value for understanding acts which express joy, pride, and happiness, which have both personal and social roles. A unique model was developed to represent these expressions of emotions in the form of body language. The data obtained herein clarify the relationship between the celebratory action and the individual and social facilitation of these acts. Additionally, to gain a meaningful coherence within our data, the stated study's objectives were followed, and the selected exploratory research design and data analysis procedures aligned with these objectives. We established a solid theoretical model which was associated with the study's objectives, methodology, and findings. In the process of data collection, appropriate behavioral coding practices were conducted with full agreement among the main coder and his affiliates.

## Statistical analyses

Following data collection, the number of single, double, team, and total celebration actions were summarized. Also, the time (in seconds) of each goal celebration was considered. Descriptive (M and SD) statistics were subjected to one- and two-way factorial analysis of variance (ANOVA) considering competitive stage (preliminary vs. final), time phase (1–15 min, 15–45 min, 45–75 min, 75–90 min, 95+ min), scoring state before and after the goal (behind, draw, lead), scoring team (home vs. away), and scorer's continent of origin (Europe, South America, or Africa) as factors in the analyses. Cohen's d coefficients were computed to estimate the standardized differences between means when effects were significant ($p < .05$).

## Results

### Descriptive data of celebration acts and their duration

To describe the goal celebration acts across the competition stage (e.g., preliminary and final), match location (e.g., home and away), time-phase (0–15, 15–45, 45–75, 75–90, 90+ minutes), scoring mode (e.g., prior to goal, after goal), and players' continent origin (Europe, Africa, Asia, South/Central and North America), the number and percent of all the celebration acts were counted and presented in their respective mode (e.g., single, double, team). The distribution of single acts is presented in Fig 2.

**Table 1. Number and % of goal celebrations in 366 games and 22 celebration modes.**

| Celebration Mode/Actions | Single | Double | Team | Total |
|---|---|---|---|---|
| Total # of actions | 14 | 4 | 4 | 22 |
| Actions performed | M = 1.99 | M = 0.31 | M = 0.94 | M = 3.25 |
| | SD = 1.55 | SD = 0.61 | SD = 0.36 | SD = 1.63 |
| | 61.23% | 9.53% | 28.92% | 100% |

The most frequent dual celebration acts following a goal were: outreaching hands (N = 123, 33.6%), hand fist (N = 109, 29.8%), appealing to audience (N = 96, 26.2%), predetermined movement (N = 91, 24.9%) and religious virtue (N = 64, 17.5%). Similar distribution for the double celebration acts is presented in Fig 3.

The most frequent double celebration act was pair movements (N = 71, 19.4%), followed by pointing toward the passer (N = 31, 8.5%). Finally, the team celebration acts' distribution is shown in Fig 4.

The most frequent and dominant team celebration act following a goal was team movements (N = 331, 90.4%); in few cases players running toward the audience (N = 9, 2.5%), and joining bench players (N = 4, 1.1%). Since some of the goal celebrations consisted of acts which comprised single, double, and team acts interactively, the distribution of all celebration acts was computed and is presented in Table 2. Along with the celebration occurrence, the mean time of the celebration acts was also computed and presented.

Of the 366 goal celebrations, the video clips of 318 (86.88%) could be used for time duration measurement. Most of the goal celebration acts (n = 196, 61.63%) were performed interactively by single and team modes. The double or single + double celebration modes were very rare. Single and team modes of celebration were also rare (6.29% and 8.8%, respectively). The single + double + team interactive mode of celebration lasted the longest (26.34 s), followed by single + team mode (25.81 s). All the interactive modes of celebration except one lasted longer (23.71

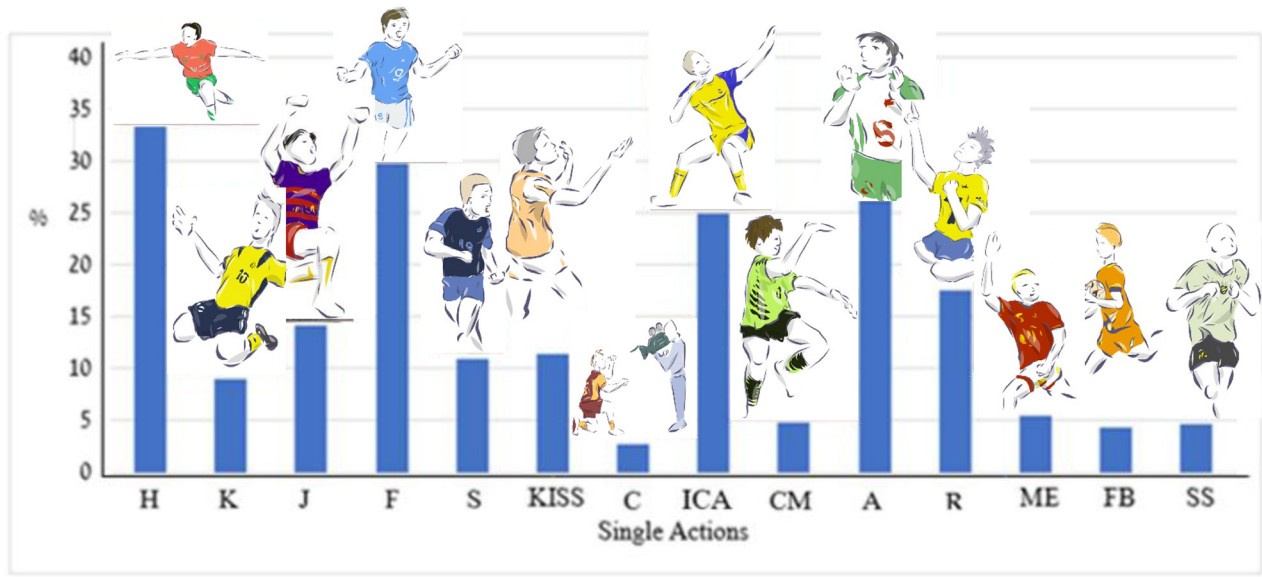

**Fig 2. Percent of single player's acts (K = 14; # of actions = 731) in 366 goal celebrations.** Index: H–Outstretched hands; K–Glitch; J–Air jump; F–Hand fist; S–Vocal scream; KISS–Emotion expression (kissing/heart); C–Approaching camera; ICA–Pre-determined movement; CM–Spontaneous movement; A–Appealing to audience; R–Religious virtue; ME–Self-aggrandizement; FB–Ball catching; SS–Tapping shirt symbol.

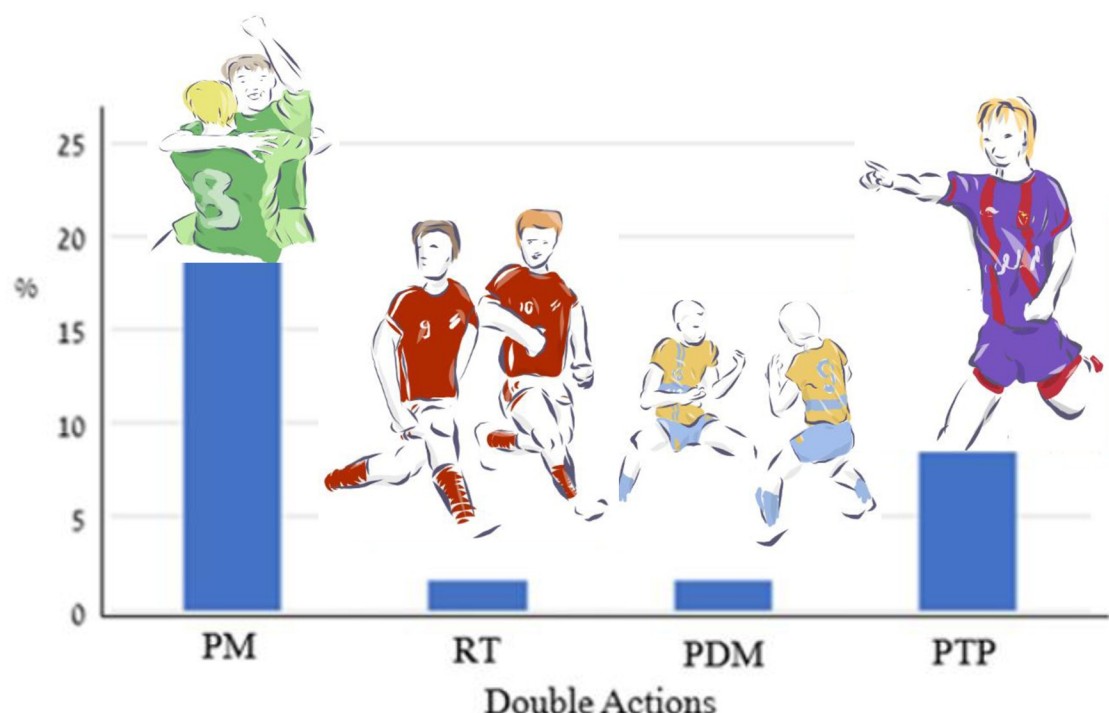

**Fig 3. Percent of double players' celebration acts (*K* = 4; # of actions = 114) in 366 goal celebrations.** Index: PM–Pair movements; RT–Running together; PDM–Predetermined movements; PTP–Pointing toward the passer.

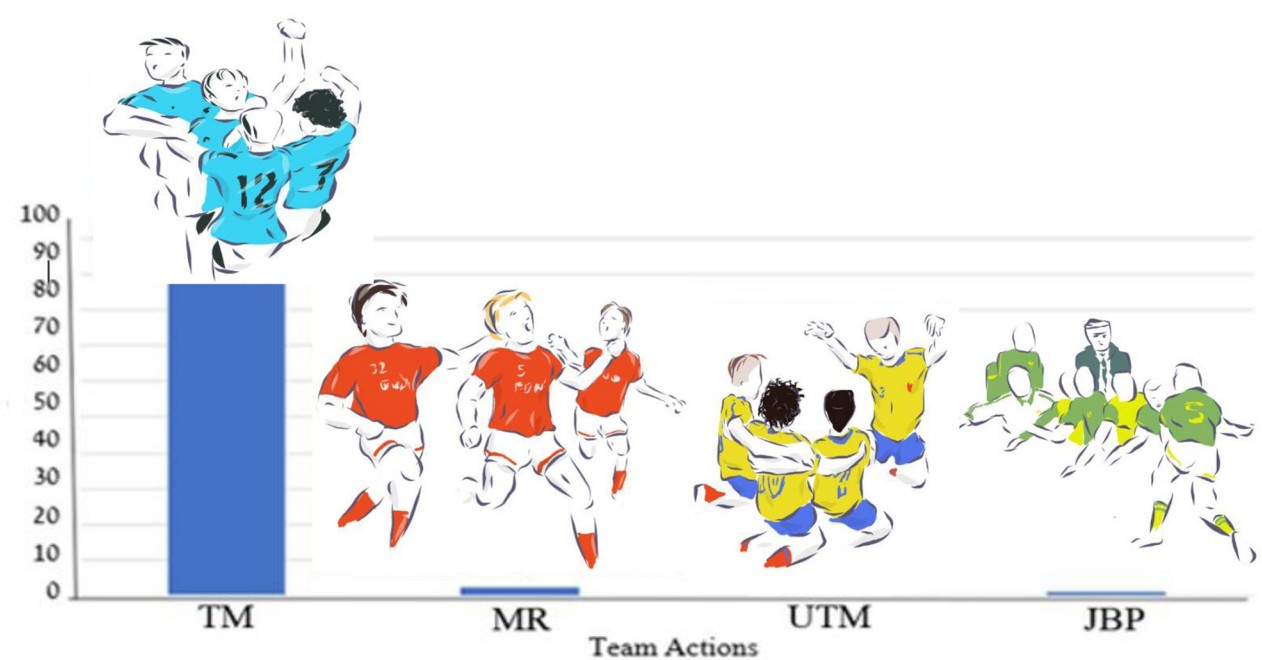

**Fig 4. Percent of team players' celebration actions (k = 4; # of actions = 345) in 366 goal celebrations.** Index: TM–Team movements; UTM–Unified team movements; MR–Players run toward audience; JBP–Join bench players.

**Table 2. Means and SD's for goal celebration time by celebration mode.**

| Celebration Mode | N | $M_t$ | $SD_t$ |
|---|---|---|---|
| Single | 20 | 22.95 | 10.59 |
| Double | 1 | 20.00 | – |
| Team | 28 | 19.25 | 6.63 |
| Single + Double | 2 | 17.50 | 3.54 |
| Single + Team | 196 | 25.81 | 10.82 |
| Double + Team | 24 | 23.71 | 8.25 |
| Single + Double + Team | 47 | 26.34 | 9.24 |
| Total | 318 | 24.90 | 10.21 |

s– 26.34 s) than the modes which were performed non-interactively (19.25 s– 22.95 s). The two cases wherein the acts were performed by a single + double mode lasted the shortest amount of time (17.50 s).

## Celebration actions and durations in the preliminary and final competition stages

The number of goal celebration acts, along with their time durations, were calculated and contrasted to each other by considering the competition stage (e.g., preliminary vs. final).

A two-way ANOVA followed by LSD post-hoc mean difference tests were performed to detect significant differences. The descriptive distribution of the goal celebration acts by celebration mode for the two competition stages is presented in Fig 5.

The ANOVA revealed non-significant ($p > .05$) differences between the preliminary and final stages on all celebration modes. The time of the celebration is presented in Fig 6.

The analysis revealed a significant competition stage effect, $F(1,319) = 193.40$, $p < .001$. The celebration lasted longer following goal scoring in the final stage than in the preliminary stage ($M = 38.69$, $SD = 13.88$ $vs$ $M = 22.09$, $SD = 6.36$, $Cohen's$ $d = 1.54$).

## Goal celebration acts and duration as a function of time phase and competition stage

Goal celebration duration was measured in 6-time phases during the matches, as follows: 1–15 min, 15–45 min, 45–75 min, 75–90 min, and 90+ min. The 45+ min time phase was deleted due to the small number of goal celebrations occurring in this time phase.

To examine the main and interaction effect of time phase (5 time intervals) and competition stage (i,e, preliminary and final), two-way factorial ANOVAs were performed for celebration duration and each of the celebration acts (e.g., single, double, team, total) separately.

The two-way ANOVA applied to the celebration duration revealed significant competition stage effect, $F(1,316) = 180.58$, $p < .001$, time phase effect, $F(4,316) = 7.53$, $p < .01$, and competition stage by time phase interaction, $F(4,316) = 7.43$, $p = .01$. Celebration duration across all time phases was longer at the final stage than at the preliminary stage ($M = 38.69$, $SD = 13.88$ $vs$. $M = 22.09$, $SD = 6.36$, $Cohen's$ $d = 1.54$). The time-phase effect revealed that celebration duration increased gradually throughout the game phases, from $M = 27.24$ s, $SD = 9.37$ during the 1-15-time phase to $M = 45.24$ s, $SD = 13.47$ during the 90+ time phase (see Fig 7). Moreover, the time phase by competitive stage interaction (see Fig 8) revealed that the celebration duration lasted the longest time when the goal was scored during the overtime phase (90 + min) of the final but not the preliminary stage ($M_{90+f} = 67.50$, $SD_{90+f} = 10.61$ vs. $M_{90+p} = 22.95$, $SD_{90+p} = 8.09$, $Cohen's$ $d = 4.72$). The time phase by competition stage interaction effect

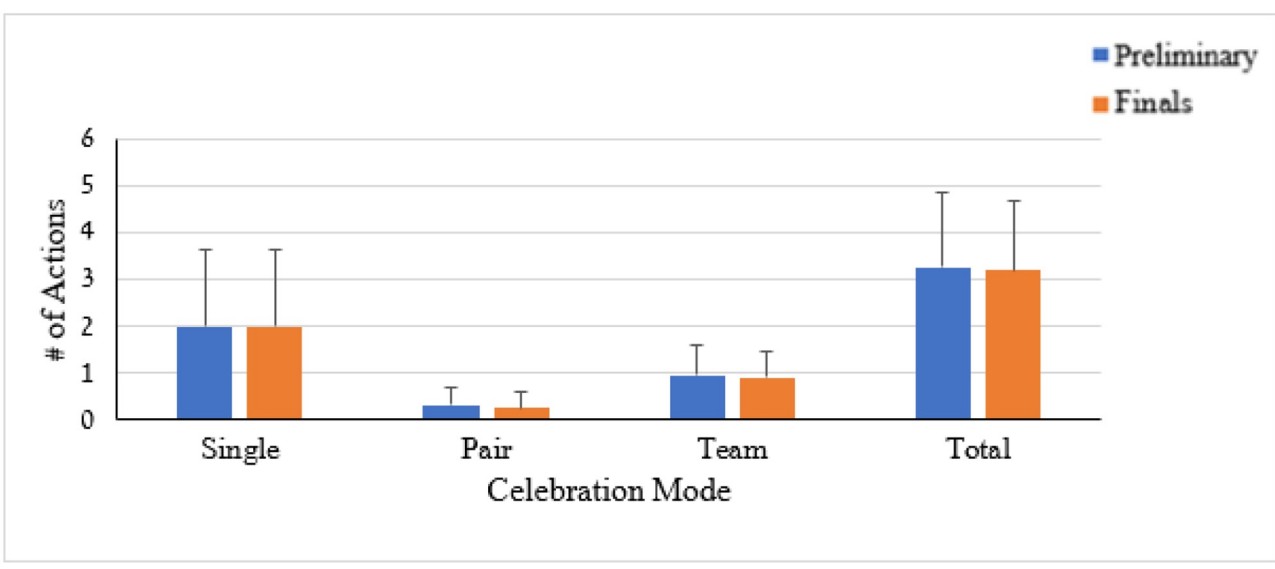

**Fig 5. Means and SD's for number of actions by celebration mode and competition stage (preliminary, final).**

on celebration time is attributed to the drastic change in celebration time during the overtime (90+ min) phase in the final competition stage.

The two-way ANOVAS pertaining to the number of acts revealed a non-significant ($p >$ .05) effect for either time phase, competition stage, or interaction. The total number of acts by the two factors is shown in Fig 9. Though non-significant, the number of celebration actions in the final stage seems to increase from the game outset to the final moments of the game but decrease during the overtime phase. In contrast, the number of celebration acts tends to decrease monotonically from the outset to end and the overtime phases of the game during the preliminary stage.

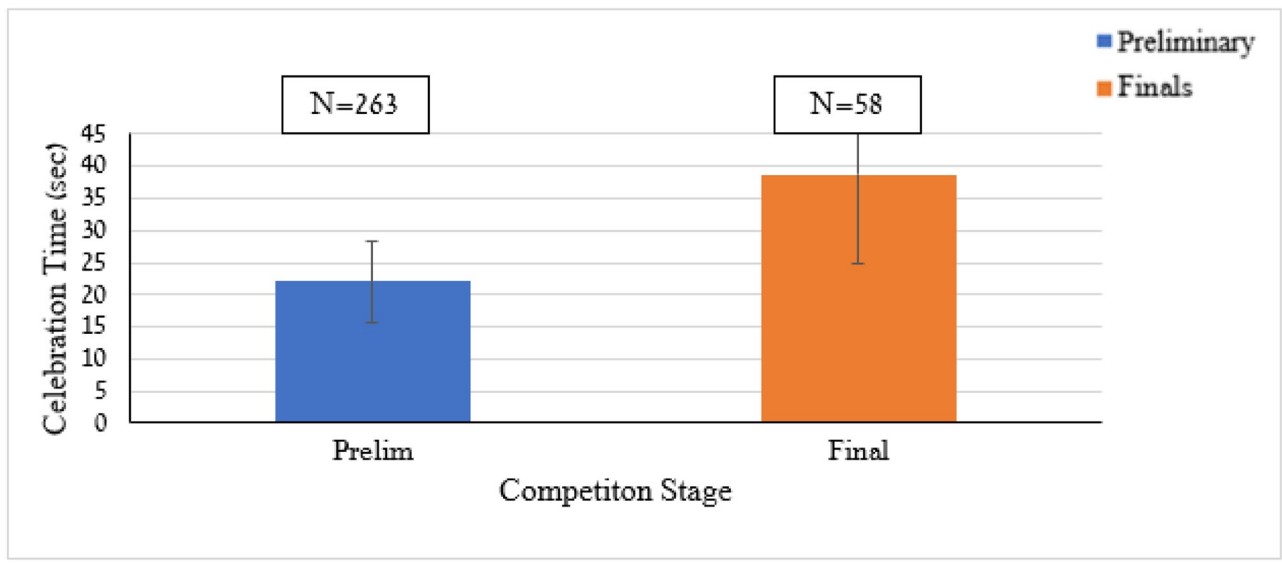

**Fig 6. Means and SD's for goal celebration time in the preliminary and final competition stages.**

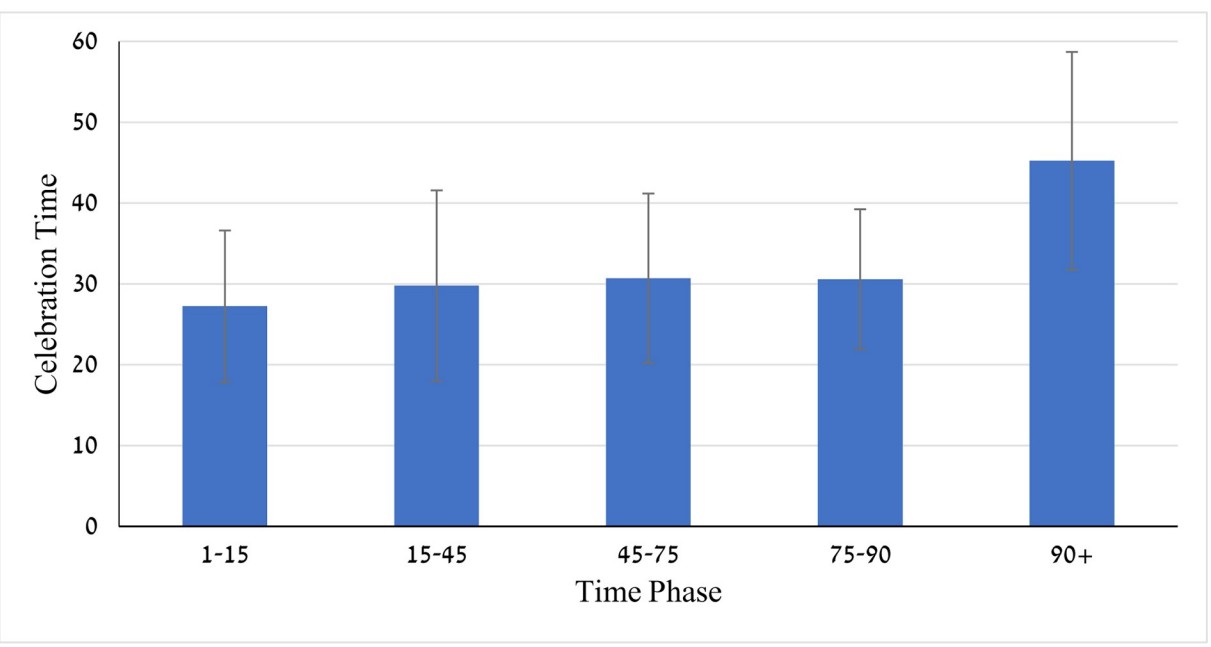

**Fig 7. Means and SDs of celebration duration through 5-time phases of the competitive matches.**

## Goal celebration acts and duration by competitive stage and scoring team (home vs. away)

Two-way ANOVA applied to the celebration time by the scoring team (e.g., home vs. away) and competitive stage (e.g., preliminary vs. final) revealed only a significant ($p < .05$) competitive stage effect. This effect is presented in Fig 6 and revealed that during the final stage the goal celebration duration is longer than in the preliminary stage.

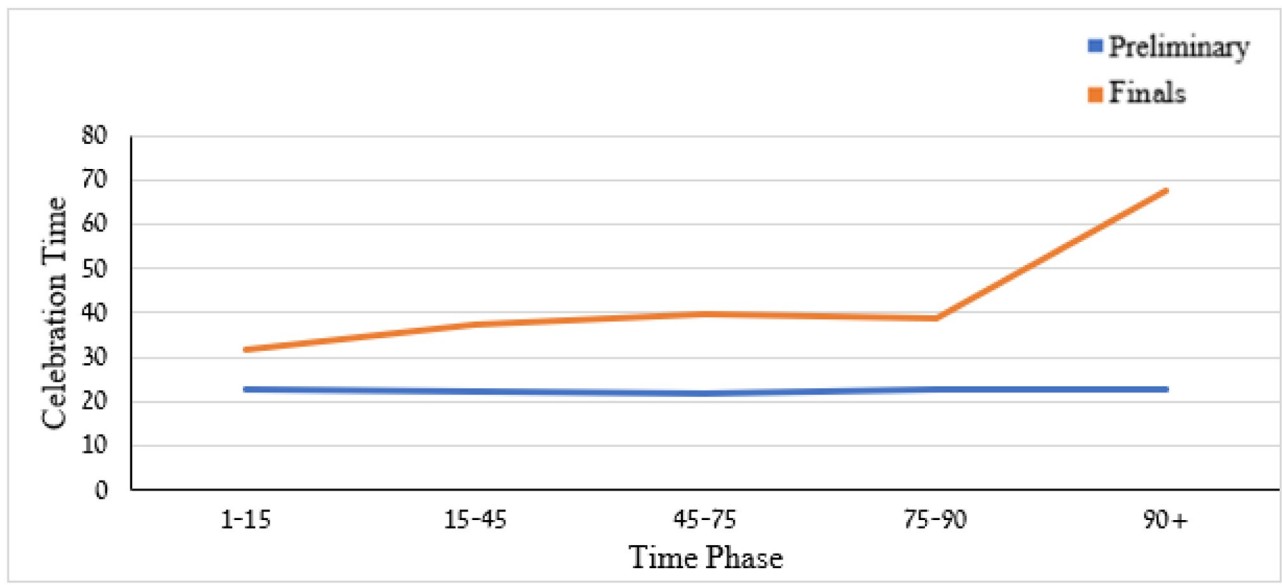

**Fig 8. Means of goal celebration duration by time phase and competition stage.**

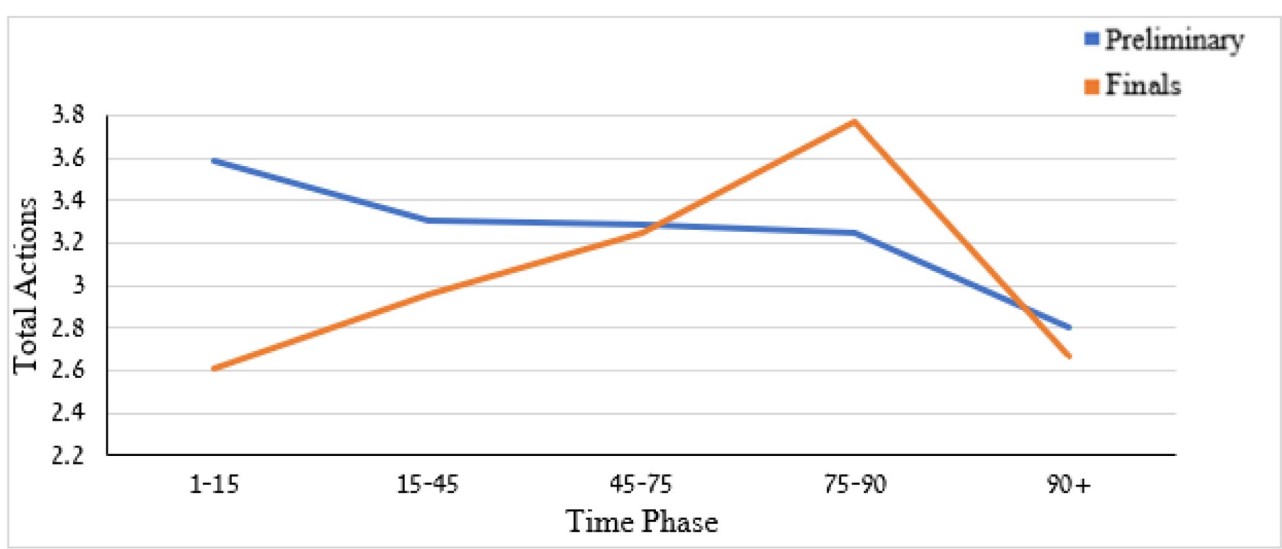

**Fig 9. Means of total actions celebration by competition stage (preliminary and final) and time phase.**

Similar analyses were performed for the single, double, team, and total number of celebrations acts separately. All the main and interaction effects resulted in a non-significant effect ($p > .05$), except the two-way scoring team by competitive stage interaction effect on the number of team acts, $F(1,362) = 3.70$, $p < .05$. This interaction is presented in Fig 10. In the preliminary stage the away team players exhibited more team celebration acts than the home team players ($M = .99$, $SD = .34$ vs $M = .92$, $SD = .37$, *Cohen's d* = 0.20). In contrast, in the final stage the home team players exhibited more team celebration acts than the away team players ($M = .96$, $SD = .39$ vs $M = .87$, $SD = .34$, *Cohen's d* = 0.25).

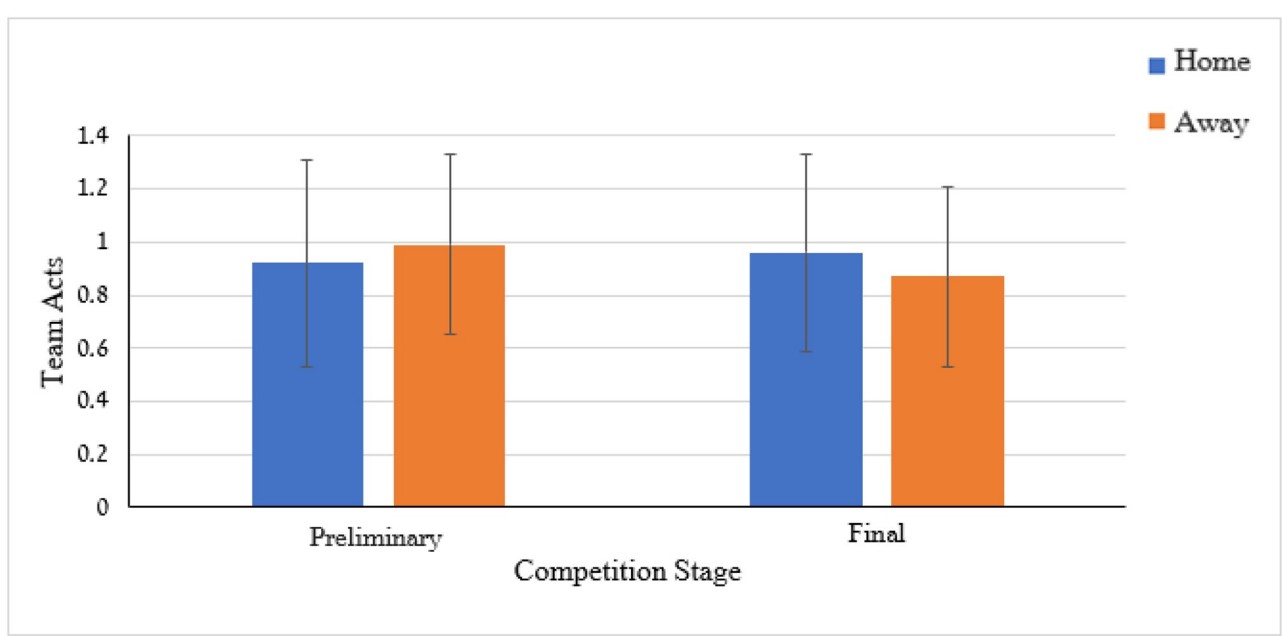

**Fig 10. Means and SDs for team celebration actions by competition stage (preliminary and final) and scoring team (home vs. away).**

## Goal celebration duration and game score status

Celebration duration was measured after scoring when the team was behind, in a draw, or leading in scoring position. One-way ANOVA considering the scoring factor revealed a significant effect for scoring status, $F(2,318) = 4.09$, $p < .02$. When scoring from a behind state, the celebration lasted a shorter time than when leading or in drawing states ($M = 22.03$ s, $SD = 9.47$ vs. $M = 25,56$ s, $SD = 10.93$ and $M = 26.27$ s, $SD = 10.18$, respectively).

Similar analysis was performed when considering the score status after the goal. The ANOVA revealed significant scoring status effect, $F(2,318) = 6.91$, $p < .001$. When the scoring team was behind, the celebration time was much shorter than leading or drawing ($M = 18.08$ s, $SD = 6.51$ vs. $M = 25.94$ s, $SD = 10.52$ and $M = 24.22$ s, $SD = 10.18$. respectively).

## Goal celebration acts and game score status

Single celebration acts were measured after scoring when the team was behind, in a draw, or leading. One-way ANOVA considering the scoring factor revealed a significant effect for scoring status, $F(2,365) = 3.19$, $p < .04$. The number of single celebration acts was higher when the score prior to the goal was a draw than when the team was behind or in lead positions ($M = 2.24$, $SD = 1.63$ vs. $M = 1.79$, $SD = 1.43$ and $M = 1.85$, $SD = 1.50$, respectively). Similar analysis pertaining to the double celebration acts revealed non-significant effect ($p > .05$). However, a significant score status was revealed for the team acts, $F(2,365) = 4.73$, $p < .01$. The number of team celebration acts was lower when the score of the team before the goal occurred was behind than when the team was in a draw or a lead position ($M = .84$, $SD = .46$ vs. $M = .95$, $SD = .39$ and $M = .99$, $SD = .23$, respectively). Consequently, the total number of celebration acts was different with respect to score status, $F(2,365) = 3.76$, $p < .02$. The number of celebration acts was the lowest when the team was behind than when it was in a draw or in a lead position ($M = 2.88$, $SD = 1.54$ vs. $M = 3.48$, $SD = 1.66$ and $M = 3.22$, $SD = 1.60$, respectively).

Considering the scoring status after the goal, the analyses pertaining to the single and the double celebration acts were non-significant ($p > .05$). However, the analysis for the teams acts was significant, $F(2,365) = 7.01$, $p < .001$. When the goal resulted in a lead position, the number of the team celebration acts was greater than when drawing or staying behind ($M = .97$, $SD = .32$ vs. $M = .91$, $SD = .40$ and $M = .71$, $SD = .53$, respectively). A similar analysis for the total number of celebration acts revealed a significant scoring status effect, $F(2,365) = 4.195$, $p < .02$. When the goal resulted in a lead position the number of the celebration acts was greater than when drawing or staying behind ($M = 3.36$, $SD = 1.64$ vs. $M = 3.09$, $SD = 1.66$ and $M = .2.46$, $SD = 1.23$, respectively).

## Celebration actions by players of various continent origins

To designate the goal celebration acts to the players' original culture, all the single, double, and team acts were divided by the players' continent origin. These data are shown in Fig 11. Celebration acts which occurred in less than 20% of the cases were designated in white. Players were assigned to one of three continents: Europe, Africa, or South America. The number of goal celebration acts for single, double, and team modes were expressed in counts and percentage. The acts which were used marginally by all the players were deleted and were not included in Fig 11. As one may notice, players of all continent origins were quite similar in celebrating in pairs (PM) and together with their teammates (TM). Pertaining to single actions, African players celebrate the goal via single acts such as outstretched hands, appealing to the audience, and religious virtue. South American players prefer single acts such as hand fists, outstretched hands, predetermined movements, appealing to the audience, and religious virtue. European players' most-used single actions are outstretched hands, hand fists, predetermined

| Continent | Single | | | | | | | | Double | Team |
|---|---|---|---|---|---|---|---|---|---|---|
| | H | J | F | S | KISS | ICA | A | R | PM | TM |
| Europe (N=219) | n=72 32.9% | n=35 16% | n=72 32.9% | n=30 13.7% | n=26 11.9% | n=48 21.9% | n=59 26.9% | n=26 11.9% | n=43 19.6% | n=199 90.9% |
| South America (N=91) | n=36 39.6% | n=15 16.5% | n=27 29.7% | n=5 5.5% | n=11 12.1% | n=31 34.1% | n=24 26.4% | n=26 28.6% | n=16 17.6% | n=82 90.1% |
| Africa (N=37) | n=12 32.4% | n=1 2.7% | n=7 18.9% | n=4 10.8% | n=3 8.1% | n=7 18.9% | n=9 24.3% | n=10 27% | n=7 18.9% | n=32 86.5% |
| Celebration Acts | | | | | | | | | | |

**Fig 11. Celebration actions' frequency by players' continent origin.**

movements, and appealing to the audience. Thus, outstretched hands and appealing to the audience are common, frequently used single celebration acts cross-culturally, while African players perform fewer air jumps, hand fists, vocal screams, expressions of emotion (kissing/heart fingers), and predetermined movements, and South Americans use fewer air jumps, vocal screams, and emotion expressions (kissing/heart fingers). Of note, Europeans refrain from religious virtue more than their African and South American counterparts.

## Discussion

The aim of this paper was to deconstruct the celebratory acts among elite professional football players in the European Champions League following scoring a goal. By analyzing the celebration from a socio-psychological perspective, we established an interactive model which is comprised of several theories. Considering the model as context dependent (e.g., preliminary and final football matches), the general contribution of the model is centered on its capacity to dismantle and to comprehend mainly the overt performances of goal celebration among football players in different stages and timing.

It is commonly agreed that goal celebration in sport is mostly spontaneous, repetitious, and manifested via vocal and bodily gestures aimed at expressing emotions. Expression of emotion is an act which signifies achieving personal and social objectives [27–29], which are largely determined by culturally driven motives [30]. We examined the goal scoring festivity in a multi-layered background by looking into single, dual, and team celebrations.

The findings revealed that the most frequent single celebration is outstretching hands followed by a fist display and appealing to the audience, suggesting there are both personal and collective festivities which correspond to the notion of personal and collective roles that emotions share (e.g., [3, 16, 20]). While the first two acts emphasize the individual aspect ('I scored

the goal, look at me!'), the latter stresses that the scorer is an integral part of a much bigger entity ('I did it for you people and the club!'). Similarly, the most frequent dual celebration act was expressed as pair players moving together as one, followed by pointing toward the passer indicating 'it isn't just me'. Both cases correspond to the newly created model (Fig 1) which guides our observations wherein overt behavior such as body gestures result from goal scoring (i.e., the stimulus). The celebration which followed evoked basic intrapersonal feelings such as cheerfulness, joy, and pride, along with interpersonal messages of loyalty, solidarity, and team cohesion, which serve cultural and interpersonal functions [2, 4, 32].

Drawing on Goffman's [23] theoretical perspective, such goal celebration ('gives-off') is a stage where the player strives to manage his impression as part of his 'front region', and performs his role for the audience. Yet such behavior is also part of a collective action performed not just by individuals or pairs. As football is a team sport, the collective–both in sporting and social acts–plays a critical role in any festivity/celebration. As Goffman states:

> 'Individuals may be bound together formally or informally into an action group in order to further like or collective ends by any means available to them. In so far as they co-operate in maintaining a given impression, using this device as a means of achieving their ends, they constitute what has here been called a team.'

[23, p. 84]

Our findings support the notion of the social role associated with expressions of emotions, in that team movements were by far more evident following goal scoring. In this respect, goal celebration emphasizes the loyalty and solidarity of both the scoring player and his teammates to the club for whom he plays. To a lesser extent, team celebrations were also joined by bench/staff members. However, it must be considered that scoring a goal is usually an act that was pre-planned by several players, and thus the celebration is an immediate response mediated by social interactions with the scorer's teammates, coaching staff, bench players, and spectators. Moreover, most of the goal celebration acts (n = 196, 61.63%) were performed interactively by singles and teammates. As Moll et al. [3] shows, and in accordance with emotional contagion postulation [8], our findings revealed that the post-performance emotions expressed by an individual were transferred spontaneously to the scorer's teammates through the intended induction of emotional states and associated behavioral attitudes [40]. A social context, such as scoring a goal, encourages the expression of emotions in people witnessing these emotions [33], who share a strong identification with the club. The expression of an emotional state in one person transfers to the experience of a comparable emotion in persons perceiving the expression, especially in spectators who identify with the action and emotions of joy, happiness, and pride [5, 6]. Within this process, emotions influence other peoples' emotions, manners, and conduct in a similar manner [3, 33].

While our analysis revealed non-significant ($p > .05$) differences between the preliminary and final stages of competition on all celebration modes, the goal celebration lasted significantly longer following goal scoring in the final stage of the tournament than in the preliminary stage. Corresponding to our model, during the final stages, both intrapersonal (pride, joy) and interpersonal (situational, contextual, and media watchers) aspects are comprised, since at this stage the knockout phase (16 teams/quarterfinal) in the tournament is decisive. In the 2018/19 season, for example, the VAR (video assisting referee) was used for the first time from that stage onwards. Additionally, our findings revealed that the celebration duration lasted the longest (by 50%) when the goal was scored during the overtime phase (90+ min) of the final competition stage, but not during the preliminary stage. The goal scored is a decisive

and imperative one because the chances are slim for the score to be overturned, due to time constraints. This makes the celebration 'sweeter' and considered to be a conclusive act.

Where home and away celebrations were considered in the current study, it was revealed that in the preliminary stage the away team players displayed more team celebration acts than the home team players. In contrast, in the final competition stage the home team players exhibited more team celebration acts than the away team players. Moreover, the number of celebration acts was the lowest when the team was behind rather than when it was in a draw or lead position. Players in a behind position are eager to continue the game, in order to score additional goals and not 'waste' time for celebration. Thus, goal celebration is expressed more frequently under a game condition where superiority over the rivals is evident, but not under a condition where superiority is not evident. Football matches must be considered as special social contexts where the spontaneous and pre-planned celebrations (fast and slow systems [12]) and expression of emotions are unique and context dependent. As such, environmental, social, and cultural issues must be taken into account when including celebrations and their related emotional expressions within this context [32].

Considering the cultural perspective, players of all continent origins were quite alike, but also unique in celebrating in pairs and together with their teammates (TM). This suggests that there are cultural differences and functions at work [2, 33]. Concerning single actions, African players celebrated the goal via single acts such as outstretched hands, appealing to the audience, and religious virtue. South American players preferred other single acts such as hand fist, outstretched hands, predetermined movement, appealing to the audience, and religious virtue, and European players mostly used single acts such as outstretched hands, hand fist, predetermined movement, and appealing to audience. Hence, outstretching hands and appealing to the audience are communal and habitually used single celebration acts cross-culturally, while African players used fewer air jump, hand fist, vocal scream, emotion expression (kissing/heart fingers), and pre-determined movement, and South Americans used less air jump, vocal scream, and motion expression (kissing/heart fingers). Additionally, European players refrained from religious virtues more than their African and South American counterparts. The act of celebration on the field brings the players' cultural background to the 'front stage' [23]. Playing in the Champions League molds players from different continents into more of a homogenized group, where they shed some of their native customs; however, during times of celebration, players tend to maintain their cultural backgrounds. An example of the homogenization of football players can be found in Acheampong's [41] study regarding the behavioral changes in African players post migration to European professional football. In Acheampong's study, he notes that African players in Europe distance themselves from their native locales after integrating into the European high-class group. However, in the context of maintaining their culture, during times of celebration players from Africa and South America demonstrate performances of religion far more than their European counterparts. This may reflect the fact that the nonreligious population of Europe is far greater than that of Africa or South America (see [42]). After all, sports cannot be disentangled from larger social practices such as religion and politics. In both Africa and South America, sport is intertwined with control of the body and space, and has become closely associated with social identity and the expression of a people's social character [43, 44]. For example, in Africa sports serve to teach moral values such as religion, and 'the maintenance and transformation of identities can be examined beautifully through sports' [44, p. 337].

The conceptual model presented in this study enables the categorization of a variety of personal and social meanings to the celebrations on the field during the most thrilling moments of the game. Such moments are habitually taken for granted by fans and scholars. As Sekerka and Fredrickson [45] assert, establishing a positive emotional climate in organizations adds to

growth and eventually better performance. Our findings illuminate how, in a team sports setting–namely football, through the process of emotional contagion, the cohesion of teammates may directly benefit from expressed positive emotions following scoring a goal.

As a final point, we strongly believe that our model can also be utilized in other team sports, such as hockey and American football. Nevertheless, there were several limitations that must be acknowledged. First, our model does not cover covert behaviors that may help shed light on both individual and team festivity. Examining the pulse, blood pressure, brain activity, and level of sweat, and respiration level are examples that may help us better understand the celebration act. Since the present data were collected during a single tournament (i.e., Champions League 2018–19), its generalization to other leagues matches (English Premier League, Bundesliga, etc.) or other tournaments (Football World Cup) must be further explored.

## Supporting information

**S1 Data.**
(XLSX)

## Acknowledgments

The authors wish to thank Art Director Yehuda Hillman for his graphic art; two anonymous reviewers for their invaluable comments, and Shaul Kashtan plus Ori Pnini from *SportsMatrix* for their expert support with the data collection process and building of the data set.

## Author Contributions

**Conceptualization:** Assaf Lev, Gershon Tenenbaum.

**Formal analysis:** Omer Eldadi, Gershon Tenenbaum.

**Investigation:** Assaf Lev, Yair Galily, Omer Eldadi, Gershon Tenenbaum.

**Methodology:** Gershon Tenenbaum.

**Project administration:** Assaf Lev.

**Software:** Omer Eldadi.

**Supervision:** Assaf Lev, Gershon Tenenbaum.

**Validation:** Yair Galily, Gershon Tenenbaum.

**Visualization:** Assaf Lev.

**Writing – original draft:** Assaf Lev, Yair Galily, Gershon Tenenbaum.

**Writing – review & editing:** Assaf Lev, Yair Galily, Gershon Tenenbaum.

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
