## [Decision Letter · Decision Letter 0]

14 Apr 2020

PONE-D-20-00271

Deconstructing celebratory acts following goal scoring among elite professional football players

PLOS ONE

Dear Dr. Lev,

Thank you for submitting your manuscript to PLOS ONE. After careful consideration, we feel that it has merit but does not fully meet PLOS ONE’s publication criteria as it currently stands. Therefore, we invite you to submit a revised version of the manuscript that addresses the points raised during the review process.

We would appreciate receiving your revised manuscript by May 23 2020 11:59PM. To enhance the reproducibility of your results, we recommend that if applicable you deposit your laboratory protocols in protocols.io, where a protocol can be assigned its own identifier (DOI) such that it can be cited independently in the future. For instructions see: http://journals.plos.org/plosone/s/submission-guidelines#loc-laboratory-protocols

We look forward to receiving your revised manuscript.

Kind regards,

Christian Stamov Roßnagel

Academic Editor

PLOS ONE

Additional Editor Comments (if provided):

Dear Dr Lev,

I truly hope all is well with you!

Your paper certainly has merit, as you can see from the attached review. Similar to the reviewer, however, I think that information is currently missing that is essential for an informed decision on your manuscript. Before we can proceed with your manuscript, I therefore ask you to address the most critical questions.

Specifically, please give more information on

1) the sample: were all matches of an entire Champions League season analysed? More information on this is needed to convince readers that there was no selection bias and that the sample is representative.

2) the coding of the celebratory behaviour. As the reviewer notes, coding the kind of whole body movements that you studied is highly challenging. From the manuscript, it cannot be judged if the person who did the coding had had sufficient training, even more so as there is a vast literature on behavioural coding that you do not refer to. Related to this, please explain why you chose not to refer to much of the relevant literature in setting up this research (see review for details).

Please respond at your next convenience, preferably before 20th April. Based on your information, we will decide without further delay on the next steps in the review process.

Happy to answer all your questions.

Best regards,

Christian.

Journal Requirements:

2. We note that Figures 2, 3, 4 includes an image of a patient / participant / in the study. 

3. Please ensure that all your conclusions are sufficiently supported by data and/or references. For example, you have stated that:

“However, in the context of maintaining their culture, during times of celebration, players from Africa and South America demonstrate performances of religion far more than their European counterparts. This is given the fact that the nonreligious population of Europe is far greater than that of Africa or South America (see World Economic Forum website, 2019).” But have not proven this link in the context of your research; thus, this should be presented as a hypothesis not as an statement.

4. Thank you for including your ethics statement:  "It was approved by the University’s ethical committee and sent only to the PLOS ONE

journal."

5. Please include your tables as part of your main manuscript and remove the individual files. Please note that supplementary tables (should remain/ be uploaded) as separate "supporting information" files

Reviewers' comments:

Reviewer's Responses to Questions

**Comments to the Author**

1. Is the manuscript technically sound, and do the data support the conclusions?

Reviewer #1: Partly

2. Has the statistical analysis been performed appropriately and rigorously? 

Reviewer #1: I Don't Know

3. Have the authors made all data underlying the findings in their manuscript fully available?

Reviewer #1: No

4. Is the manuscript presented in an intelligible fashion and written in standard English?

Reviewer #1: Yes

5. Review Comments to the Author

Reviewer #1: The present research attempts to deconstruct the celebratory behavior of soccer players based on televised recordings of the 2018-2019 Champions League season.

I found several things to like about the present report. Most importantly, this is an understudied topic and I believe the immense documentation of sports events is well suited to advance the empirical study of e.g. emotions and nonverbal behavior in an ecologically valid way. However, I do have some substantial reservations regarding the report at present. Potentially some of these issues could be addressed in a revision.

Major comments

1) The introduction could be more focused. Please make sure to be explicit about the rationale of the paper and the aim(s) of this research. Also, if you do decide to conduct Null-Hypothesis-Significance-Testing, make sure to report hypotheses based on sound theory.

2) An important issue in this paper regards sampling celebratory behavior in soccer. Where all matches of an entire season of the Champions League analyzed? More information on this is needed to convince readers that there was no selection bias in the study and that the sample studied is representative considering the claims the authors would like to make.

3) The most important problem with this paper at present is the coding of the celebratory behavior. This is not a trivial issue as coding whole body movements in non-standardized real-life contexts that were filmed for different durations and from different perspectives is highly challenging. Hence, it is insufficient to state that one person was trained to code the body language. There is a vast literature base on behavioral coding in psychology. But the authors do not refer to any of this literature and I am skeptical if the method used in this research is reliable. The authors need to substantially elaborate on this. This information is essential for me to make an informed recommendation concerning publication of this research.

3b) A related point. The authors seem to have missed a lot of relevant literature in setting up this research, especially on nonverbal behavior coding and nonverbal behavior in sports.

Minor comments:

1) At the end of the introduction you state that you analyzed 125 matches in the European Premier League during 2019. Champions League seems the less confusing description here.

2) It’s Kahneman not Khanman. By the way, dual-process theorizing is quite common in sports and the authors might want to incorporate some of the dual-process work in sports into their manuscript.

3) Is the historical information in the section “the Champions League framework” necessary? This section seems not very important towards the papers aims.

4) This is not a very important point, but is it correct that “The UEFA Champions League final is the most watched annual sporting event in the world”? I thought this was the Super Bowl.

5) I don’t understand how 79 teams (from 54 European associations) can be correct

6. PLOS authors have the option to publish the peer review history of their article (what does this mean?). If published, this will include your full peer review and any attached files.

Reviewer #1: No

---

## [Author Response · Author response to Decision Letter 0]

14 May 2020

Given the fact that the response letter to reviewers is formatted in tables, the system does not allow it to be pasted in this box. Response letter to reviewers is attached as file in this submission.

---

## [Decision Letter · Decision Letter 1]

22 Jul 2020

PONE-D-20-00271R1

Deconstructing celebratory acts following goal scoring among elite professional football players

PLOS ONE

Dear Dr. Lev,

Thank you for submitting your manuscript to PLOS ONE. After careful consideration, we feel that it has merit but does not fully meet PLOS ONE’s publication criteria as it currently stands. Therefore, we invite you to submit a revised version of the manuscript that addresses the points raised during the review process.

As one minor revision, to avoid a potential misunderstanding, please acknowledge in the manuscript that a limitation of the study was that it utilized an ad-hoc coding procedure and we do not know how reliable the coding was. On p. 10 you write "According to Dael, Mortillaro, and Scherer (2012) ... there is no consensus on a reliable coding system", which may be taken to imply that intercoder reliability may not be determined; however, Dael and colleagues did just that in the quoted paper. The second minor change concerns the original data that should be in English. Please change this.

We look forward to receiving your revised manuscript.

Kind regards,

Sara Fuentes Perez, PhD

Staff editor

On behalf of:

Christian Stamov Roβnagel 

Academic Editor

PLOS ONE

Reviewers' comments:

Reviewer's Responses to Questions

**Comments to the Author**

1. If the authors have adequately addressed your comments raised in a previous round of review and you feel that this manuscript is now acceptable for publication, you may indicate that here to bypass the “Comments to the Author” section, enter your conflict of interest statement in the “Confidential to Editor” section, and submit your "Accept" recommendation.

Reviewer #1: All comments have been addressed

2. Is the manuscript technically sound, and do the data support the conclusions?

Reviewer #1: Yes

3. Has the statistical analysis been performed appropriately and rigorously? 

Reviewer #1: Yes

4. Have the authors made all data underlying the findings in their manuscript fully available?

Reviewer #1: No

5. Is the manuscript presented in an intelligible fashion and written in standard English?

Reviewer #1: Yes

6. Review Comments to the Author

Reviewer #1: I reviewed the initial submission of this manuscript and found much to like about the study. I think the authors have responded well and professionally to my comments. This has resulted in an improved manuscript. As I consider this to be interesting and novel data I think this manuscript should be published. I do have some minor suggestions.

My only more substantial concern still regards the coding of the video material. As I have done this a lot (or tried to do this on videos from sports recordings) I know that this is a problematic issue. I think the authors should acknowledge in the manuscript that a limitation of the study was that it utilized an ad-hoc coding procedure and we do not know how reliable the coding was. Also, the authors could give the video clips to a second coder and asses intercoder reliability with the first coder. Would this be possible?

Two additional minor suggestions:

1) It would be helpful if the original data file would be in English. Please change this.

2) Would it be possible to upload the videos of the celbrations (e.g. OSF page)? This would help other researchers to build on this work. Or do additional coding of the material.

7. PLOS authors have the option to publish the peer review history of their article (what does this mean?). If published, this will include your full peer review and any attached files.

Reviewer #1: No

---

## [Author Response · Author response to Decision Letter 1]

17 Aug 2020

This document has been attached as a separate file as the system does not support the necessary format.

---

## [Editor Report · Decision Letter 2]

24 Aug 2020

Deconstructing celebratory acts following goal scoring among elite professional football players

PONE-D-20-00271R2

Dear Dr. Lev,

Thank you for your revised version. I've really enjoyed working with you during the review process!

We’re pleased to inform you that your manuscript has been judged scientifically suitable for publication and will be formally accepted for publication once it meets all outstanding technical requirements.

Kind regards,

Christian Stamov Roßnagel

Academic Editor

PLOS ONE

---

## [Editor Report · Acceptance letter]

27 Aug 2020

PONE-D-20-00271R2 

Deconstructing celebratory acts following goal scoring among elite professional football players 

Dear Dr. Lev:

I'm pleased to inform you that your manuscript has been deemed suitable for publication in PLOS ONE. Congratulations! Your manuscript is now with our production department. 

Kind regards, 

on behalf of

Mr Christian Stamov Roßnagel 

Academic Editor

PLOS ONE